# Comparison of Culture- and Quantitative PCR-Based Indicators of Antibiotic Resistance in Wastewater, Recycled Water, and Tap Water

**DOI:** 10.3390/ijerph16214217

**Published:** 2019-10-30

**Authors:** Jaqueline Rocha, Telma Fernandes, Maria V. Riquelme, Ni Zhu, Amy Pruden, Célia M. Manaia

**Affiliations:** 1CBQF—Centro de Biotecnologia e Química Fina—Laboratório Associado, Escola Superior de Biotecnologia, Universidade Católica Portuguesa, Rua de Diogo Botelho 1327, 4169-005 Porto, Portugal; jrocha@porto.ucp.pt (J.R.); telma.7919@gmail.com (T.F.); 2Department of Civil and Environmental Engineering, Virginia Tech, Blacksburg, VA 24061, USAniz@vt.edu (N.Z.)

**Keywords:** antibiotic resistance monitoring, antibiotic resistant coliforms, water quality

## Abstract

Standardized methods are needed to support monitoring of antibiotic resistance in environmental samples. Culture-based methods target species of human-health relevance, while the direct quantification of antibiotic resistance genes (ARGs) measures the antibiotic resistance potential in the microbial community. This study compared measurements of tetracycline-, sulphonamide-, and cefotaxime-resistant presumptive total and fecal coliforms and presumptive enterococci versus a suite of ARGs quantified by quantitative polymerase chain reaction (qPCR) across waste-, recycled-, tap-, and freshwater. Cross-laboratory comparison of results involved measurements on samples collected and analysed in the US and Portugal. The same DNA extracts analysed in the US and Portugal produced comparable qPCR results (variation <28%), except for *bla*_OXA-1_ gene (0%–57%). Presumptive total and fecal coliforms and cefotaxime-resistant total coliforms strongly correlated with *bla*_CTX-M_ and *intI*1 (0.725 ≤ R^2^ ≤ 0.762; *p* < 0.0001). Further, presumptive total and fecal coliforms correlated with the *Escherichia coli*-specific biomarkers, *gadAB,* and *uidA*, suggesting that both methods captured fecal-sourced bacteria. The genes encoding resistance to sulphonamides (*sul1* and *sul2*) were the most abundant, followed by genes encoding resistance to tetracyclines (*tet*(A) and *tet*(O)) and β-lactams (*bla*_OXA-1_ and_,_
*bla*_CTX-M_), which was in agreement with the culture-based enumerations. The findings can help inform future application of methods being considered for international antibiotic resistance surveillance in the environment.

## 1. Introduction

Antibiotic resistance and the associated increasing failure of therapeutic and life-saving drugs has been recognized to be a major threat to human health in the 21st Century [1,2,3]. In recent years, there is growing interest in implementing monitoring campaigns to better understand sources and pathways by which antibiotic resistance may originate and spread in the general environment, e.g., water, sewage, manure, soil, sediments, and dust [4,5,6]. However, standardized approaches are needed to promote comparability of the data and to address major questions such as the role of various management practices, country to country and region to region, in exacerbating or attenuating antibiotic resistance levels.

Phenotypic antibiotic resistance can be assessed based on culture-based methods or inferred from culture-independent approaches [7,8]. Culture-based methods, often used to assess antibiotic resistance in clinical pathogens, may be of limited value for environmental bacteria, given that most bacteria do not grow well under typical laboratory conditions [7,8,9] and that intrinsic antibiotic resistance phenotypes, common in environmental bacteria, can confound results. Notably, non-culturable bacteria may play an important role in the spread of antibiotic resistance as a reservoir of antibiotic resistance genes (ARGs). Methods based on the direct analysis of environmental DNA have, thus, became popular for quantifying and assessing diversity of ARGs [8,10]. Quantitative PCR (qPCR) is a targeted method that uses DNA primers specific for given genes or gene families, permitting the detection and quantification of a priori selected target genes [8,11]. While newer metagenomics approaches offer promise in non-target ARG profiling, the overall advantage of qPCR remains in more precise quantification of target gene populations and on the possible ability to detect low-abundance genes [8]. qPCR has been used to quantify ARGs in different environments, e.g. clinical samples, groundwater, wastewater, manure, and soil [11,12,13,14,15,16,17,18,19,20]. Nevertheless, qPCR is not exempt from biases, which include the inability to ascertain gene expression (when DNA based), the potential for qPCR inhibition, and the inability to directly discriminate between extracellular and intracellular DNA [13,21,22,23,24].

Traditionally, microbiological water quality is assessed based on coliforms analyses, a method still used widely throughout the world [25,26,27]. Coliforms are thus a target of interest for standardized antibiotic resistance monitoring (Marano et al., in preparation; [28]). Major advantages include feasibility of implementation, low technical requirements, and potential for global data comparability. Among the limitations, neglecting the non-culturable populations, lack of representativeness of the full microbial community, and lack of bench-marking against culture-independent methods remain as challenges.

The purpose of the present study was to comprehensively assess both coliform-based and qPCR-based measures of antibiotic resistance across a spectrum of water types: domestic wastewater, recycled water, river water, and drinking water. In particular, we focused on phenotypic measures of tetracycline, sulphonamide, and 3rd-generation cephalosporin resistance among coliform and enterococci bacteria along with whole-community quantification of ARGs pertaining to corresponding classes: *tet*(A), *tet*(O), *sul1*, *sul2*, *bla*_OXA-1,_ and *bla*_CTX-M_. Further, the class 1 integrase gene, *intI1*, was quantified as an indicator of anthropogenic sources of acquired antibiotic resistance and potential to be mobilized [29]. International inter-laboratory comparison of the various methods provides insight into information gained by various approaches and can help inform standardized methods for environmental monitoring in the future.

## 2. Materials and Methods

### 2.1. Water Systems and Sampling

Samples were collected in the US and in Portugal. US samples included wastewater influent (RWW), secondary wastewater treatment effluent (sTWW), tertiary wastewater treatment final effluent (tTWW), river water, tap water (TW), and distributed reclaimed (i.e., recycled) water (SRWDS). Each water sample was collected on three different dates (Appendix A). The US wastewater treatment plant (WWTP) employed primary and secondary treatments operating with conventional activated sludge and a tertiary treatment based on UV disinfection. River water samples were collected approximately 10 km downstream from the WWTP discharge point. The drinking water was a chloraminated municipal water supply collected from a laboratory tap equipped with a granular activated carbon filter, while the simulated reclaimed water distribution system was chlorinated. Tap water and simulated water distribution system water samples were tested for chlorine residual levels before downstream experiments, in order to neutralize disinfectants [30], however no chlorine was detected in any of the samples.

Portugal (PT) wastewater samples included the influent (RWW) and secondary wastewater treatment effluent (sTWW) of a WWTP, collected during two sampling campaigns (Appendix A). The PT WWTP analysed employed primary and secondary treatment, in accordance with conventional activated sludge.

### 2.2. Culture-Based Methods

For US samples, microbiological water quality was assessed based on presumptive total and fecal coliforms and presumptive enterococci [31], using the membrane filtration method. Membrane fecal coliform medium (mFC agar, Difco, Chicago, USA) was used to enumerate total and fecal coliforms and m-Enterococcus (mEnt) agar medium (Sigma-Aldrich, St Luis, USA) to enumerate enterococci. Antibiotic resistant subpopulations were quantified on these culture media supplemented with one of the following antibiotics: cefotaxime (CTX, 4 mg/L; Sigma-Aldrich, St Luis, USA), tetracycline (TET, 16 mg/L; Sigma-Aldrich, St Luis, USA), or sulfamethoxazole (SMX, 350 mg/L; Sigma-Aldrich, St Luis, USA). The antibiotic concentrations were selected based on the Clinical and Laboratory Standards Institute (CLSI) minimal inhibitory concentrations for *Enterobacteriaceae*, as previously described [32,33,34]. For PT samples, presumptive total and fecal coliforms were enumerated as described above, as well as the cefotaxime-resistant populations. Volumes ranging from 1 to 100 mL of water or of the adequate serial decimal dilutions were filtered through cellulose nitrate membranes (0.22 μm porosity; Sartorius Stedim Biotech, Göttingen, Germany), placed on the adequate culture medium and incubated at 37 °C for 24 h for total and fecal coliforms and for 48 h for enterococci. Experiments were done in triplicate.

### 2.3. Culture-Independent Methods

Water samples were filtered in triplicate for each sampling date through polycarbonate membranes (sterile 0.22 μm porosity, Merck Millipore, USA). Filter membranes were stored at −80 °C until DNA extraction using the FastDNA™ SPIN KIT (MP Biomedicals LCC, Illkirch, France) for US samples or PowerWater® DNA Isolation Kit (MO BIO Laboratories Inc., Carlsbad, USA) for PT samples, according to manufacturer instructions. The extracted DNA was quantified using the Qubit 3.0 Fluorometer (ThermoFisher Scientific, Waltham, USA). Real-time PCR was used to quantify 16S rRNA, 23S rRNA (specific gene for enterococci) [35], *uid*A, and *gad**AB* (specific for *Escherichia coli)* [36,37], *bla*_CTX – M,_
*bla*_OXA-1_ (resistance determinants against β-lactams) [38,39], *sul*1 and *sul*2, (resistance determinants against sulphonamides) [40], *intI*1 (the class 1 integrase gene and indicator of anthropogenic sources of resistance and gene mobilization potential) [41], and *tet*(A), and *tet*(O) (resistance determinants against tetracyclines) [42,43] genes. Data was expressed as absolute abundance (gene copy number/mL of sample) or as relative abundance or prevalence (gene copy number/ 16S rRNA gene copy number). The qPCR assays were performed in a Bio-Rad Real-Time PCR Analysis Software (Biorad, Richmond, CA, USA) in the US and in a StepOneTM Real-Time PCR System (Life Technologies, Carlsbad, CA, USA) in PT. The qPCR conditions used are listed in Appendix A. For each sample, either for US or for PT samples, three DNA extracts were analysed using the Standard Curve method as described in Brankatschk and collaborators [44]. The variability of qPCR between the two laboratories, using in-house laboratory qPCR standards, reagents, and protocols (Appendix A), was tested on the same DNA extracts, analysed in US and Portugal facilities. With this aim, the DNA extracts were shipped refrigerated on ice in a styrofoam box from the US to Portugal, where they were immediately stored at −20 °C. DNA extract concentrations were analysed again prior to qPCR assays, and the values were consistent with those determined in the US, suggesting that any potential DNA degradation was minimal. In both the US and Portugal facilities, identical criteria for analysis and interpretation of results were followed, although distinct qPCR protocols were an intentional factor in this study [45]. Possible qPCR inhibition was assessed using serial dilutions of the samples’ DNA extracts and samples spiked with a known amount of the target gene.

### 2.4. Statistical analyses

Culture-based and qPCR results were analysed based on one-way analysis of variance (ANOVA) and Tukey’s post-hoc tests in order to assess statistically significant differences (*p* < 0.01) using GraphPad Prism 7.00 software. Normality and homoscedasticity of data were verified using the Shapiro–Wilk and Levene’s tests, respectively. When normality of data was not verified, the Kruskal–Wallis test was carried out alternatively to one-way ANOVA. In such cases, the Mann–Whitney test was used to detect differences. Significance level across the study was set at *p* < 0.01. In addition, the relationship between the culturable bacteria (log CFUs/mL) and the gene absolute abundance values (log gene copy number/mL of sample) was assessed via the linear regression method using GraphPad Prism 7.00 software. This analysis was based on 13 samples (US: wastewater influent (n = 3), secondary wastewater treatment effluent (n = 3), and tertiary wastewater treatment final effluent (n = 3) and PT: wastewater influent (n = 2) and secondary wastewater treatment effluent (n = 2). Linear regressions were performed between the genes 16S rRNA, *gadAB*, *uid*A, *bla*_OXA-1_, *bla*_CTX-M_, *sul1*, *sul2, tet*(O), and *intI*1 and the culture media mFC and mFC supplemented with cefotaxime.

## 3. Results

### 3.1. Culture-Based and qPCR Determinations

Presumptive total coliforms ranged from 1.7 to 6.3 log CFU/mL, with the highest values in the influent wastewater and the lowest in the river water (Figure 1). Presumptive fecal coliforms, represented approximately 4 to 89% of the presumptive total coliforms and were detected mostly in the influent wastewater and in the secondary treatment effluent, at around 5.5 and 1.9 log CFU/mL, respectively in US and 5.6 and 5.0 log-units in PT samples. Presumptive enterococci presented similar distribution and abundance as presumptive fecal coliforms. Presumptive total and fecal coliforms and presumptive enterococci were not detected in tap water or in distributed reclaimed water. Antibiotic resistant populations across all samples analysed were <44% for sulfamethoxazole, <17% for tetracycline, and <4.8% for cefotaxime of the presumptive total coliforms; <84% for sulfamethoxazole, <49% for tetracycline, and <1.5% for cefotaxime of the presumptive fecal coliforms; and <126% for sulfamethoxazole, <241% for tetracycline, and <133% for cefotaxime of the presumptive enterococci (Appendix A). These high percentages (>100%) are likely an artifact, due to the loss of culture medium selectivity for enterococci, with the overgrowth of other bacterial groups on mEnterococci agar.

The 16S rRNA gene served as a proxy measurement of total bacteria, ranging from 8.2–5.3 log gene copies per mL of sample. The 16S rRNA gene absolute abundance values were observed to be the highest in PT influent wastewater samples and the lowest in tap water (Figure 2). Biomarker genes for *Escherichia coli*
*(E. coli)*, *gadAB,* and *uid*A, and for enterococci, 23S rRNA, were only detected in the influent wastewater, secondary effluent, and tertiary treatment effluent in the US samples. In PT wastewater samples, the genes *gadAB* and *uidA* were detected in the influent and secondary treatment effluent. Most of the time, the secondary treatment led to significant decreases in the abundance of these genes in US (*p* < 0.01), but not in PT wastewater samples (Appendix A). Presumptive total coliforms were detected in river water, while the genes *gadAB* and *uid*A could not be quantified in the same samples (Figure 1 and Figure 2). In general, as expected, the influent wastewater presented the highest absolute abundance of targeted ARGs (Figure 2). The genes encoding resistance to sulfonamides (*sul1* and *sul2*) and the class I integron integrase (*intI*1) were the most abundant and prevalent in all samples, followed by genes encoding resistance to tetracyclines (*tet*(A) and *tet*(O)) and β-lactams (*bla*_OXA-1_ and *bla*_CTX-M_) (Figure 2 and Appendix A). Since the genes encoding resistance to tetracycline (*tet*(A) and *tet*(O)) were detected in the secondary treatment effluent and in the tertiary treatment effluent, but not in the influent wastewater, PCR inhibition was assessed. To test this, influent wastewater DNA extracts where it was not possible to quantify those genes were diluted or spiked with exogenous DNA template to confirm functionality of the assay. These assays did not lead to amplicon detection or permit the quantification of the spiked exogenous DNA, discarding the hypothesis of qPCR inhibition in those extracts. In general, the pattern of gene abundance per volume of water was similar of that of gene relative abundance (per 16S rRNA gene) (Figure 2). The simulated reclaimed water distribution system samples deviated most strongly from this trend, where a high 16S rRNA gene load likely contributed to low relative abundance determination in spite of the high load of ~5 log-units of gene copy per mL, of *tet*(A), *sul1*, and *sul2* genes.

### 3.2. Molecular Biomarkers and Culture-Based Methods

A linear regression model was used to infer possible correlations between presumptive total and fecal coliforms and 16S rRNA, *gadAB*, *uid*A, *bla*_OXA-1_, *bla*_CTX-M_, *tet*(O), *sul1*, *sul2,* and *intI*1 genes. The strongest correlations, with R^2^ > 0.725 and *p* < 0.0001, were observed between the *E. coli-*specific genes, *uid*A and *gadAB*, the β-lactams encoding gene *bla*_CTX-M_ and the class I integron integrase *intI1* and the presumptive total coliforms, presumptive fecal coliforms, and cefotaxime-resistant presumptive total coliforms (Table 1). Correlations between fecal antibiotic resistant populations and the analysed genes were not among the strongest. The correlations between presumptive total, fecal, and cefotaxime-resistant total coliforms and the genes 16S rRNA, *bla*_OXA-1_, *tet(O)*, *sul1,* and *sul2* were not statistically significant.

### 3.3. Quantitative PCR Inter-Laboratory Comparability

Quantitative PCR analyses were performed for the 16S rRNA, *gadAB*, *bla*_OXA-1_, *sul1,* and *sul2* genes in two laboratories (US and PT) using the same DNA extracts, handled by the same operators. The following genes were selected for the inter-laboratory comparison, in part due to logistical reasons associated with the timeline of available protocols during the international exchange portion of the study: i) among the genes analysed in the US laboratory (16S rRNA, 23S rRNA, *gadAB*, *sul1*, *sul2*, *tet*(A), *tet*(O), and *bla*_OXA-1_), in the first stage, enterobacteria and related genes were observed to be better biomarkers for water quality than enterococci and related genes and ii) for some of these genes, it was possible to compare distinct protocols already tested in each laboratory, fulfilling the laboratories cross-comparison study aim. For these analyses, distinct equipment, protocols, references, and reagents were used (Appendix A). Major discrepancies in the results corresponded to situations in which a specific gene was detected only by one of the laboratories. For example, *gadAB* in most of the samples in US or *sul2* in wastewater influent 3A (RWW3A) in PT (Figure 3). Distinct limits of quantification might be a contributing factor (10 gene copies in the US and 43 gene copies in PT for *gadAB* gene) in some situations, while in other situations it may be related to shifts that could have occurred during DNA shipment (*sul2* gene, quantification in river water 3A DNA extract). When quantification was possible in both laboratories, discrepancies between the two labs ranged from 0.0 log-units, for 16S rRNA gene in tap water 3A (TW3A) sample, to 2.7 log-units, for *bla*_OXA-1_ gene quantification in wastewater influent 3A (RWW3A) sample. In general, the PT lab yielded absolute abundance values that were equal or greater than those reported by the US lab for influent wastewater but tended to report lower values than the US lab for the other “cleaner” samples (Figure 3). With the exception of the discrepancy observed for the *bla*_OXA-1_ gene (variation between 0% and 57%), the gene quantification coefficient of variation was generally satisfactory between labs, ranging from 0% to 28% for the remaining analysed genes.

## 4. Discussion

In contrast with readily detected target biomarker genes in the US tertiary treatment effluent, culture-based methods consistently yielded CFUs/mL close to the LOQ: −1.7 log CFUs/mL for presumptive fecal coliforms and −1.7 log CFUs/mL for enterococci (Figure 1). However, presumptive total coliforms were detected in river water, while the *gadAB* and *uid*A genes could not be quantified in the same samples (Figure 1 and Figure 2). The absolute abundance of antibiotic resistance genes analysed was in agreement with the culture-based enumerations, where there were generally fewer cefotaxime-resistant CFUs than sulfamethoxazole- or tetracycline-resistant CFUs (Figure 1). The fact that the qPCR inhibition hypothesis was not confirmed in wastewater influent for the detection of *tet*(A) and *tet*(O) genes might suggest that these genes might be present in the influent wastewater samples at very low absolute abundance and were somehow enriched during secondary treatment.

In recent years, there has been growing interest in environmental surveillance of the levels of antibiotics, antibiotic resistant populations and ARGs, and of bacterial lineages found in wastewater treatment plants [10,17,44,45,46]. However, there is still a lack in knowledge regarding possible biomarkers that would be amenable for routine, accessible, and standardized monitoring targets and methodologies. To better understand how the occurrence of the various targets of this study relate to each other, a linear regression model was used to infer about possible correlations between presumptive total and fecal coliforms and 16S rRNA, *gadAB*, *uid*A, *bla*_OXA-1_, *bla*_CTX-M_, *tet*(O), *sul1*, *sul2,* and *intI*1 genes. These targets were selected based on the following key attributes: enterobacteria are common fecal indicators of water quality, the 16S rRNA gene corresponds to total bacteria, *gadAB* and *uid*A are genes specific to *E. coli*, the *intI*1 gene is an indicator of anthropogenic pollution and is associated with mobility of multi-antibiotic resistance, and the remaining genes as encoding antibiotic resistance commonly spread in wastewater environments, associated to pathogenic and commensal bacteria in humans [47,48,49,50]. The strongest correlations observed between the *E. coli*-specific genes (*uid*A and *gadAB*) and the presumptive total fecal and cefotaxime-resistant total coliforms confirm previous studies that demonstrate the usefulness of housekeeping genes *gadAB* or *uid*A to predict coliform abundance [36,51,52]. Also, the strong correlation between the *intI*1 gene and the presumptive total and fecal coliforms populations confirmed the role of *intI1* as a proxy for human-driven pollution sources [29]. The strong correlation of the *bla*_CTX-M_ gene with presumptive coliforms and total cefotaxime-resistant coliforms abundance suggests that coliforms could be major carriers of this resistance gene or that both coliforms and the antibiotic resistance gene are being discharged from a common source and with similar persistence rates in the water environments. However, such inferences would require confirmatory studies. The widespread occurrence of *intI*1 and *bla*_CTX-M_ genes in *Enterobacteriaceae* (e.g. *E. coli* and *K. pneumoniae*) has been demonstrated [53,54,55], which might explain the strong correlations observed between these genes and presumptive total and fecal coliforms populations observed here. Interestingly, correlations between antibiotic resistant fecal populations and the analysed genes, were not among the strongest. This observation is in line with different possible explanations: i) that in contrast with other *Enterobacteriaceae* that comprise total coliforms, *E. coli* do not proliferate in water, ii) that antibiotic-resistant *E. coli* may have a lower fitness to grow in the culture medium used than other coliforms, or iii) that it grows on that medium but not all cells form the typical blue colonies in the presence of antibiotics, due to metabolic pathway regulation. The observed correlations are in any case relevant findings considering that coliforms are frequently carriers of antibiotic resistance genes and associated with human bacterial infections. Correlations among the targets of interest in this study suggest the possibility of simplified monitoring schemes in which a target can be selected to represent potential for all of the other targets to be present. Such simplified methods are needed particularly in world regions where monitoring is needed, but laboratories are not readily available or are poorly equipped.

Rocha and colleagues [45] evaluated the variability of qPCR procedures for quantification of ARGs using the same DNA extracts and qPCR protocols between five different laboratories. Although in our study the qPCR protocols were not the same in both laboratories, our results are in agreement with those of Rocha et al. [45] in which the coefficient of variation observed ranged from 3% to 28%. Although this range of variation was determined for quantifications performed in distinct laboratories, most of the times the difference of samples Ct values between the quantifications performed in both laboratories was <3.3 cycles (corresponding to 1 log of difference between samples) suggesting that the differences in gene copy number observed might be due to the number of copies established in each laboratory. The present study was more realistic in that distinct qPCR standards, protocols, and equipment were applied, as would be the case in published literature from different labs. These results are promising, suggesting that qPCR could provide standardized quantification of target ARGs and other relevant genes in labs for global monitoring.

## 5. Conclusions

These findings aid in identifying simplified monitoring schemes for environmental sources of antibiotic resistance that are suitable for a wide range of water types and feasible for low-tech labs across the globe. Comparison of gene quantification between the two laboratories indicated that qPCR can yield consistent and highly quantitative results, even when different reagents, protocols, and equipment are used. Regression analyses comparing the abundance values of cultivable bacteria and *gadAB*, *uidA*, *bla*_CTX-M,_ and *intI*1 indicated that these genes may be suitable biomarkers for total coliforms, beta-lactam resistant subpopulations, and their possible role in the dissemination of antibiotic resistance in the environment.

## Figures and Tables

**Figure 1 ijerph-16-04217-f001:**
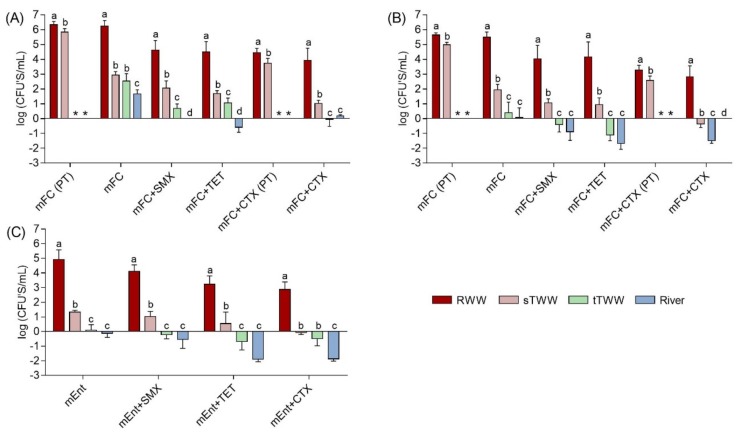
Colony forming units (CFUs) of (**A**) presumptive total coliforms, (**B**) presumptive fecal coliforms, and (**C**) presumptive enterococci expressed as log CFU /mL of sample. Bacteria were cultured on membrane fecal coliform (mFC) and m-Enterococcus (mEnt) media, with and without antibiotics: cefotaxime (CTX, 4 mg/L), tetracycline (TET, 16 mg/L), or sulfamethoxazole (SMX, 350 mg/L). Results were compared for four water types RWW: wastewater treatment plant influent, sTWW: wastewater collected after secondary wastewater treatment, tTWW: wastewater collected after UV disinfection wastewater treatment, and river water. CFUs were below the limit of quantification (1 CFU/100 mL, corresponding to −2 log units) for tap water and simulated reclaimed water distribution system water samples. a, b, c, and d indicate significantly different groups comparing the different types of water (*p* < 0.01). PT refers to data obtained of samples collected in Portugal, the other data presented refer to data obtained of samples collected in United States. * -refers to data not determined.

**Figure 2 ijerph-16-04217-f002:**
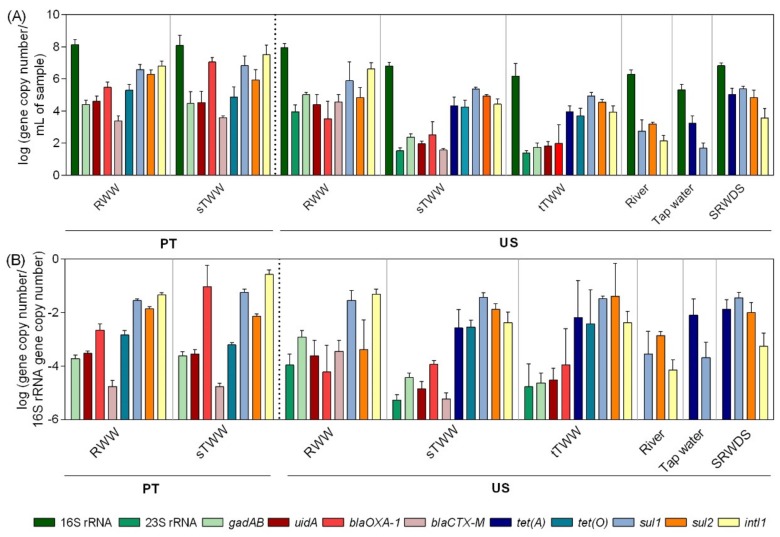
Genes absolute and relative abundance values across the water samples analysed.(**A**) Absolute abundance (gene copy number/mL of sample) and (**B**) relative abundance values (gene copy number/16S rRNA gene copy number) of the analysed genes in six water systems. RWW: wastewater treatment plant influent, sTWW: wastewater collected after secondary wastewater treatment, tTWW: wastewater collected after UV disinfection wastewater treatment, river water, tap water, and SRWDS: simulated reclaimed water distribution system. PT refers to data obtained from samples collected in Portugal, the other data presented refer to data obtained of samples collected in the US. Error bars represent the standard deviation (n = 3 and n = 2 independent samples, for US and PT, respectively). In PT samples, the genes 23S rRNA and *tet*(A) were not analysed. In US samples, the absence of the genes *bla*_OXA-1_, *tet*(A), *tet*(O), *sul2,* and *intI*1 is due to their quantification below the limit of detection while the absence of the genes 23S rRNA, *gadAB*, *uidA,* and *bla*_CTX-M_ was due to genes quantification below the limit of quantification.

**Figure 3 ijerph-16-04217-f003:**
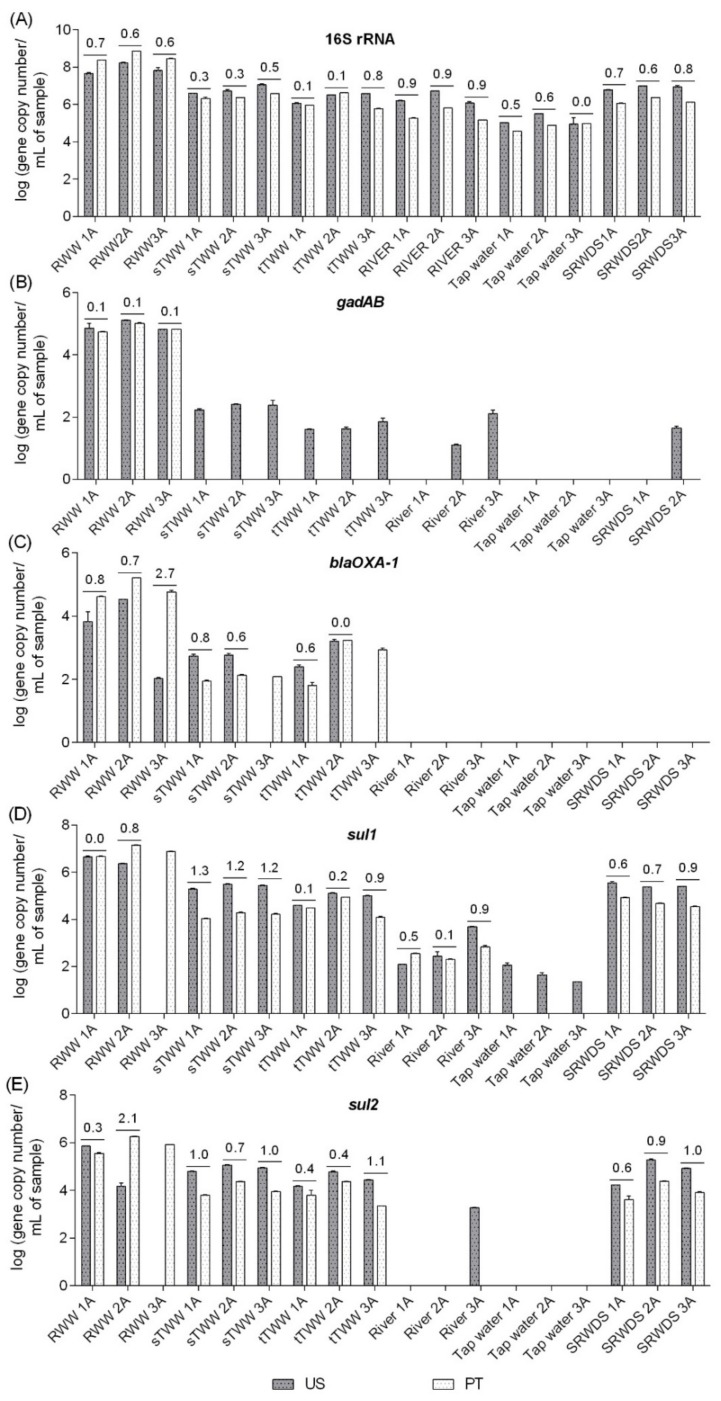
Inter-laboratory comparison of absolute abundance (gene copy number/mL of sample) measurements of the target genes in the six water types. RWW: wastewater treatment plant influent, sTWW: wastewater collected after secondary wastewater treatment, tTWW: wastewater collected after UV disinfection wastewater treatment;, river water, tap water, and SRWDS: simulated reclaimed water distribution system. PT refers to genes quantification performed in Portugal, the other data presented refer to genes quantification performed in the US. The lack of gene quantification (LOQ) performed in Portugal for some samples was due to quantification below the limit of detection and the lack of gene quantification performed in US for some samples was due to quantification below the limit of detection (*bla*_OXA-1_ and *sul2*) or due to quantification below the limit of quantification (*gadAB*). The absolute difference values in the mean quantification of technical triplicates are indicated above the bars, whenever the target was quantifiable in both labs. The gene copy numbers considered as LOQ were 100 vs. 402 for 16S rRNA gene, 10 vs. 43 for *gadAB*, 10 vs. 38 for *bla*_OXA-1_, 10 vs. 96 for *sul1,* and 100 vs. 47 for *sul2*, respectively, in the US and in PT. The quantitative polymerase chain reaction (qPCR) conditions used in both laboratories for each gene quantification are presented in the Appendix A.

**Table 1 ijerph-16-04217-t001:** Linear regression coefficients comparing target genes and colony forming units on mFC medium with and without cefotaxime (CTX, 4 mg/L). n = 13 for these analyses, including wastewater treatment plant influent (n = 3), wastewater collected after secondary wastewater treatment (n = 3), and wastewater collected after UV disinfection wastewater treatment (n = 3) samples collected in US and wastewater treatment plant influent (n = 2) and wastewater collected after secondary wastewater treatment (n = 2) samples collected in PT. TC—presumptive total coliforms and, FC—presumptive fecal coliforms.

	mFC (TC)	mFC (FC)	mFC+CTX (TC)	mFC+CTX (FC)
16S rRNA	0.399	0.375	0.331	0.000
*gadAB*	0.868*	0.861*	0.764*	0.016
*uidA*	0.805*	0.795*	0.922*	0.567
*bla* _OXA-1_	0.327	0.298	0.279	0.055
*bla* _CTX-M_	0.750*	0.761*	0.738*	0.048
*tet*(O)	0.494	0.489	0.521	0.333
*sul1*	0.247	0.220	0.157	0.037
*sul2*	0.157	0.149	0.151	0.003
*intI*1	0.762*	0.750*	0.725*	0.000

* statistically significant (*p* < 0.0001).

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
