# Peer review of "Comparison of Culture- and Quantitative PCR-Based Indicators of Antibiotic Resistance in Wastewater, Recycled Water, and Tap Water"

_ijerph, 2019, doi:10.3390/ijerph16214217_

Round 1

Reviewer 1 Report

The paper by Rocha et al. presents some data associated with spreading antibiotic resistance and the possibility of comparison of obtained results between laboratories. To do it, the Authors make similar analyses in two laboratories, one in the United States and the second in Portugal. Though the idea of the study was interesting and worthy of analyzing, the manuscript has methodological limitations and drawbacks, and in present form can not be published in IJERPH.

Major comments

The aim of the study was inter-laboratory assessing of culturable indicators counts, the quantity of genes specific for these indicators and antibiotic resistance genes. The information should help researcher to standardize methods for environmental monitoring. In my opinion, the aim of the study was not reached. Only part of the study was common for the United States and Portugal, but not all ( e.g. growing culturable bacteria on media with tetracycline and sulfamethoxazole) but all of these data are present in figures and in the text of the manuscript, what make the manuscript unclear. In my opinion, the Authors should remove data, which are not comparable between countries from the manuscript. In the case of genes, pool of studied genes was the same for the US and PT but the comparison between the laboratories was determined only for few of them. I believe, it could be related to limited volume of isolated  DNA but I do not  understand why the Authors did not choose genes for which statistically significant liner coefficients were obtained.

Minor comments

The section of M&M needs improvement, specifically description of sampling sites. At the moment, without table S1, is not clear what samples were collected in the US and what in PT. The samplings section should be split for two paragraphs. There is no reason why 16SrRNA, gadAB, blaoxa-1, sul1 and sul2 were chosen to inter - laboratory comparison. Why parametric ANOVA was used instead of non-parametric test like Kruskal-Wallis test? The counts of bacteria do not have usually normal distribution. Abbreviation TC is used for both total coli as well as for tetracycline. Please, replace abbreviation of tetracycline by new one.

Reviewer 2 Report

This is one of the best written and more scientifically sound papers I have seen on your Journal.  It is clear, very easy to follow and organize.  On top of that, the problem they address is a ubiquitous problem across the globe.  Water quality and antibiotic resistance go hand to hand with public health.  In addition, although not all labs have a PCR, currently this technology is more affordable and accessible to more and more labs even in disadvantage Countries.  This research put into perspective the importance of water quality and antibiotic resistance and the accessibility of this methodology to protect public health across thee globe.

Reviewer 3 Report

This is an interesting manuscript about the comparison of culture-based methods target species of human-health relevance and the quantification of antibiotic resistance genes (ARGs) by qPCR. Moreover, the ARGs quantification by qPCR was also compared between two different laboratories. As the authors expressed in the discussion, the comparison of the qPCR using different equipment and protocols is more realistic so the results could help to standardize the qPCR to quantify ARGs among different laboratories. Although the manuscript does not report novel techniques or approaches, the research topic is important, the data obtained is interesting and the manuscript is well written, easy to follow by the readers. Overall, I think the results are valuable enough to be published in International Journal of Environmental Research and Public Health. However, before its acceptance, I would like to discuss the following issues with the authors:

The title “Inter-laboratory comparison of culture- and quantitative PCR-based indicators of antibiotic resistance in wastewater, recycled water, and tap water” does not totally reflect what was done in the study, as the inter-laboratory comparison was done exclusively for qPCR assays and not for culture techniques. I would recommend to change the title for one more appropriated. L82: In the text, you exposed that the samples were collected in “three different dates”, but in the table S1 there are much more dates. Clarify this inconsistency, please. L101-103: Which were the criteria applied to select these antibiotic concentrations? Did you follow any guideline? L155-156: How do you explain that the percentages for resistant enterococci populations were higher than 100%? L235: Could you specify in the manuscript the conditions of the shipment? Did not you add any internal control to know the degradation of the DNA during the shipment? This is my main concern of the study because you cannot be completely sure if the coefficient of variation was due to the difference of the applied protocols or because DNA was damaged before being analyzed for the 2nd

Round 2

Reviewer 1 Report

The authors have made good improvements to the manuscript. I am satisfied that the authors have addressed all of the Reviewers’ questions and/or concerns.
I recommend accepting the manuscript for publication.